# Bedrock radioactivity influences the rate and spectrum of mutation

Nathanaëlle Saclier[1][†]*, Patrick Chardon[2], Florian Malard[1], Lara Konecny-Dupré[1], David Eme[1][‡], Arnaud Bellec[1,3], Vincent Breton[2], Laurent Duret[4], Tristan Lefebure[1]*, Christophe J Douady[1,5]

[1]Univ Lyon, Université Claude Bernard Lyon 1, CNRS UMR 5023, ENTPE, Laboratoire d'Ecologie des Hydrosystèmes Naturels et Anthropisés, Villeurbanne, France; [2]LPC, Université Clermont Auvergne, CNRS/IN2P3 UMR6533, Clermont-Ferrand, France; [3]Univ Lyon, Université Jean Moulin Lyon 3, CNRS UMR 5600 Environnement Ville Société, Lyon, France; [4]Univ Lyon, Université Claude Bernard Lyon 1, CNRS UMR 5558, Laboratoire de Biométrie et Biologie Evolutive, Villeurbanne, France; [5]Institut Universitaire de France, Paris, France

**Abstract** All organisms on Earth are exposed to low doses of natural radioactivity but some habitats are more radioactive than others. Yet, documenting the influence of natural radioactivity on the evolution of biodiversity is challenging. Here, we addressed whether organisms living in naturally more radioactive habitats accumulate more mutations across generations using 14 species of waterlice living in subterranean habitats with contrasted levels of radioactivity. We found that the mitochondrial and nuclear mutation rates across a waterlouse species' genome increased on average by 60% and 30%, respectively, when radioactivity increased by a factor of three. We also found a positive correlation between the level of radioactivity and the probability of G to T (and complementary C to A) mutations, a hallmark of oxidative stress. We conclude that even low doses of natural bedrock radioactivity influence the mutation rate possibly through the accumulation of oxidative damage, in particular in the mitochondrial genome.

*For correspondence:
nathanaelle.saclier@gmail.com
(NS);
tristan.lefebure@univ-lyon1.fr (TL)

Present address: [†]ISEM, CNRS, Univ. Montpellier, IRD, EPHE, Montpellier, France; [‡]IFREMER, unité Ecologie et Modeles pour l'Halieutique, Nantes, France

Competing interests: The authors declare that no competing interests exist.

## Introduction

Natural radioactivity is the main natural source of exposure to ionizing radiations on Earth. Natural radioactivity is generated by cosmic radiation or by radionuclides released from the bedrock. While levels of cosmic radiation fluctuates over time due to cosmic events such as supernovae or solar flares, bedrock radioactivity remained mainly stable until 2 billion years ago, when it began to slowly decrease (*Karam and Leslie, 2005*). Bedrock radioactivity depends on the nature of the rocks which extensively varies spatially (e.g. *Ielsch et al., 2017*). While few extremely naturally radioactive sites such as the India Kerala and Iranian Ramsar region have been monitored for their impact on the human mutation rate (*Forster et al., 2002*; *Masoomi et al., 2006*) or on plant physiology (*Saghirzadeh et al., 2008*), the influence of regional variation in baseline natural radioactivity on the evolution of biodiversity is still unknown (*Møller and Mousseau, 2013*).

Radioactivity can impact species' molecular evolution by modulating the rate at which different types of mutation appear and accumulate. Ionizing radiations damage DNA by breaking the DNA sugar-phosphate backbone (*Hoeijmakers, 2001*; *van Gent et al., 2001*). Alternatively, ionizing radiations can trigger the formation of reactive oxygen species (ROS) directly inside cells through the radiolysis of water (*Wallace, 1998*; *Ward, 1988*). ROS being also mutagenic (*Barja, 2002*), ionizing radiations are direct and indirect mutagens. The most deleterious and studied mutations generated by radioactivity are DNA double-strand breaks (DSBs). Two repair systems are able to manage double-strand breaks: homologous recombination (HR) and non-homologous end-joining (NHEJ). HR

uses an homologous sequence to revert the damage, but is only used at specific cell cycles when an undamaged sister chromatid is available (*Karran, 2000*). Otherwise, NHEJ is preferentially used. As NHEJ does not use a template-strand to reconstruct the missing genetic information, it only restores the continuity of the DNA molecule to the price of frequent deletions and chromosomal exchanges. *Rothkamm et al., 2001* showed that 50% of the double-strand breaks were misrejoined after a strong irradiation (80 Gy). Numerous studies demonstrate that an exposure to radiation produces chromosomal abnormalities (*Dikomey et al., 2000*; *Loucas et al., 2004*; *Hande et al., 2005*), deletions (*Jostes et al., 1994*; *Huo et al., 2001*; *Adewoye et al., 2015*; *Allegrucci et al., 2015*), and point mutations (*Huo et al., 2001*; *Winegar et al., 1994*; *Forster et al., 2002*; *Barber et al., 2002*). Radiation-induced chromosomal abnormalities and deletions have been thoroughly studied because of their frequent deleterious impact. The impact of radiation on point mutations has received less attention.

While the mutational impact of exposure to high doses of radioactivity is well characterized (*Dubrova et al., 1996*; *Ziegler et al., 1993*), the exposure to low doses of radioactivity is poorly known. Some authors (*Tubiana et al., 2006*) propose that DNA repair and apoptosis may completely counteract the effect of ionizing radiations for doses below 0.1 Gy, suggesting that low doses have no biological impact. Indeed, in vitro experiments on mammalian cells show that repair systems are more efficient to repair DSBs at low dose than at higher dose (*Boucher et al., 2004*, 0.05 vs 3.5 Gy/min). However, exposure to an even lower dose of ionizing radiation (less than 0.1 cGy/min) increases the number of mutants in mammals (*Vilenchik and Knudson, 2000*; *Hooker et al., 2004*) suggesting that repair systems are not activated at very low radiation doses. When repair systems are not activated and DNA damage accumulates, cells tend to die by apoptosis, preventing the transmission of radioactivity-induced mutations to the next cell generation (*Rothkamm et al., 2001*; *Collis et al., 2004*). Unrepaired DSBs are the main cause of radioactivity-induced cell apoptosis during mitosis (*Krueger et al., 2007*). As point mutations will not lead to cell death like DSBs, this type of radioactivity-induced mutations could stay unrepaired and be transmitted to the next generation. Exposure to low doses of radioactivity also induces an adaptive response: exposed cells are resistant to a following higher dose of radiation (see *Rigaud and Moustacchi, 1996*). An early or chronic exposure to low doses of radiation may strengthen the antioxidant defense and reduce the sensitivity to radioactivity-induced ROS. While this effect is well demonstrated for preventing DSBs, its efficiency to reduce point mutations is less certain (*Rigaud et al., 1995*). Moreover, the adaptive response seems to only protect the nuclear genome while the mitochondrial one may not benefit from it (*Jarrett and Boulton, 2005*).

The short-term impact of high doses of radiation is indisputable, but the long-term impact of exposure to low doses is unknown. Some studies found an increase in the number of mutations in the offspring of exposed people (*Dubrova et al., 1996*; *Forster et al., 2002*), while others studies found the opposite (*Satoh et al., 1996*; *Czeizel et al., 1991*), leaving open the question of the transmission of mutations generated by low doses of radioactivity. This lack of knowledge is likely contributed to by three factors. First, studies are mainly focused on the health effect of low dose of radiation and not on their long-term mutational impact (e.g. *Beir, 2006*; *Tubiana et al., 2006*). Second, most of the literature focuses on unrepaired double-strand breaks which are highly deleterious and are de facto not transmitted to the next generation. Third, studying the long-term mutational impact of natural radioactivity is challenging because it raises a number of methodological difficulties. On the one hand, experimentally exposing multiple generations of multicellular organisms to low doses of radiation would require years of experimentation and complex experimental controls. On the other hand, the main obstacles to in naturae studies are the organisms' mobility, which prevents certainty that a population was exposed to the same natural radioactivity for many generations, and confounding factors such as ultraviolet radiation from the sun.

Here, we overcome these difficulties by coupling in situ radioactivity characterizations with the distinctive bio-ecological characteristics of subterranean waterlice within a phylogenetic comparative framework. Subterranean waterlice are never exposed to UV radiation, live in contrasted bedrock set-ups and have very limited dispersal capacity (*Eme et al., 2018*), allowing us to make the assumption that different species have persisted in different but nearly constant radioactive habitats for numerous generations. To test the impact of radioactivity on the transmission of point mutations, we estimated the long-term rate of neutral mutations in the nuclear and in the mitochondrial genome independently.

## Results and discussion

In order to build a robust and powerful comparative design aimed for testing the influence of natural radioactivity on the mutation rate, we first prospected for closely related subterranean species living in contrasted radioactive set-ups. Using the map of bedrock uranium content in France (*Ielsch et al., 2017*), we prospected areas with low and high radioactivity. From this large survey (58 sites with waterlice), we selected 14 sites with contrasted levels of α radioactivity which were inhabited by closely related groundwater waterlice species. We paid special attention that α radioactivity differed at least by a factor of three between two habitats, each containing a closely related species (*Figure 1*, *Supplementary file 1*). On average low level of radioactivity was around 0.357 Bq/g of dry sediment and high level around 1.259 Bq/g of dry sediment. Based on transcriptome sequencing and de novo assembly, we used a phylogenetic approach to estimate nuclear and mitochondrial substitution rates (i.e. the rate of mutations which are fixed). While experimental approaches allow to measure the impact of radioactivity on somatic mutations or on the transmission of mutations across few generations, this phylogenetic approach allows us to measure the impact of natural radiation on the germinal mutation rate over the course of a species history.

Using the 14 selected species and locations, we tested whether there was a significant positive relationship between natural radioactivity and the long-term mutation rate. The latter was estimated using the synonymous substitution rate ($d_S$) calculated on the terminal branches of the phylogenetic tree tracing the history of these 14 species. The $d_S$ is the rate at which silent mutations accumulate in protein coding genes and, when calculated using many different loci and in the absence of a strong synonymous codon usage bias (see methods), is an estimator of the average mutation rate across a

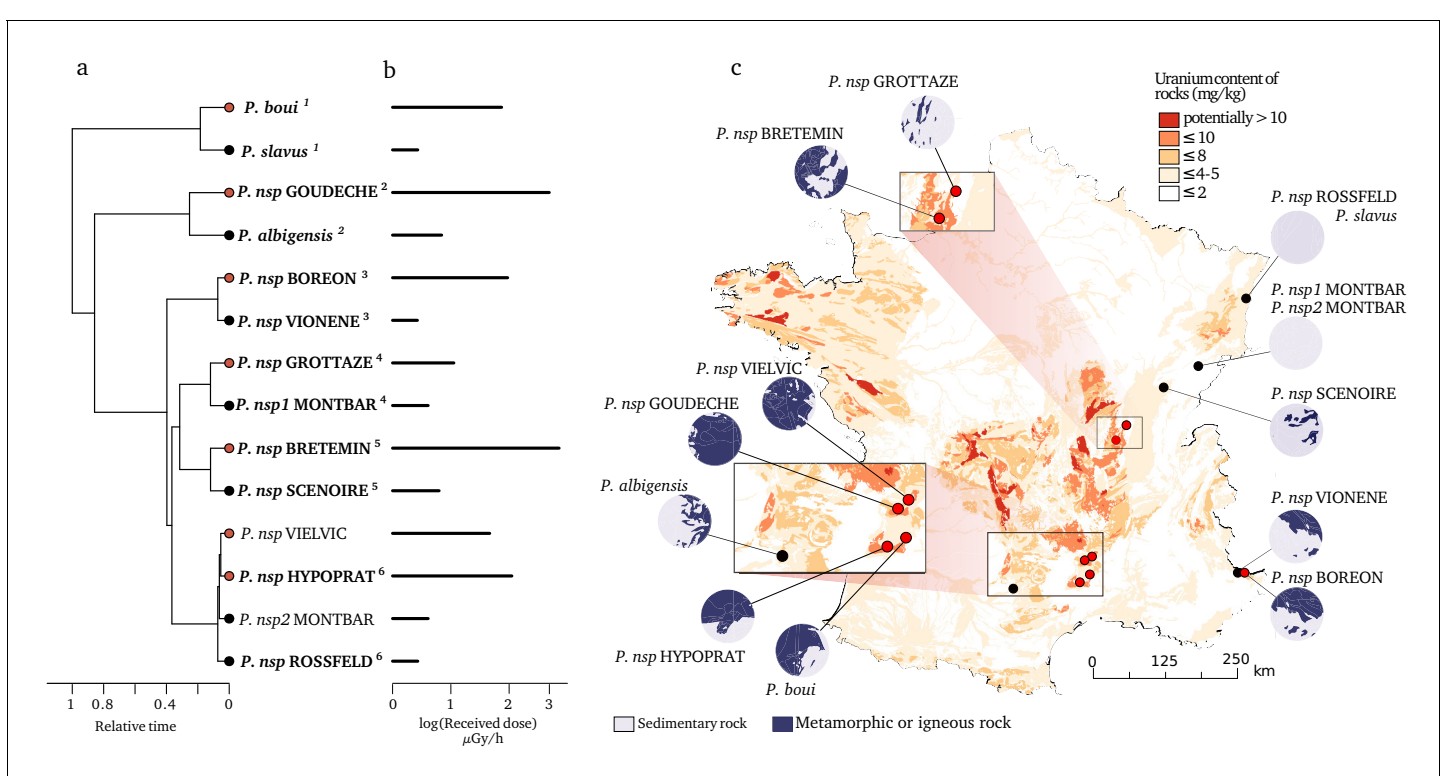

**Figure 1.** Species and locations selected to study the impact of bedrock radioactivity on the mutation rate and spectrum. Fourteen species with contrasted bedrock radioactivity exposure were selected (black dot: low exposure, red dot: high exposure). Based on their phylogenetic history (**a**), we further selected six monophyletic pairs of closely related species to compare their mutational spectrum (Vielvic and Montbar are excluded because of unresolved phylogeny, pairs are indicated using superscript numbers). Received dose of radioactivity was determined from measurements of radioactivity in the sediments of the sampled sites (**b**). For each site, the areal proportion of low-radioactivity sedimentary rocks and high-radioactivity metamorphic and igneous rocks in a radius of 15 km around the sampling (λ15) site is represented with circles next to the map (**c**).

The online version of this article includes the following figure supplement(s) for figure 1:

**Figure supplement 1.** Phylogeny of the 14 species used to compute synonymous substitution rates.

species' genome (*Kimura, 1983*). To be comparable across species, these $d_S$ has to be divided by the time over which it is measured. To achieve that, we used the software CoEvol (*Lartillot and Poujol, 2011*) which models directly a synonymous substitution rate relative to the root age ($d_S$/ra). We computed $d_S$/ra using 769 one-to-one nuclear and 13 mitochondrial orthologous protein-coding genes shared by all 14 species. At each sampling site, we measured the global α radioactivity and the activity of all radio-elements in the sediment. The analysis of the composition in radionuclides at each site reveals that two sites (BRETEMIN and BOREON) show a disruption of the secular equilibrium in the U-238 chain. This suggests that nearby industrial activities (e.g. lead mines) have modified the natural radioactivity at these two sites. As these industrial activities are very recent (since 1950), their impact on the substitution rate, which is measured on a much longer time scale, is unlikely. We therefore did not use these two sites to test the correlation between $d_S$ and any site-specific radioactivity measurement (however, see next paragraph for a regional measurement). The $d_S$/ra is positively correlated with the α radioactivity in the nuclear genome as well as in the mitochondrial genome (*Table 1*; *Figure 2*). A linear model predicts a $d_S$/ra increase of 31.8% in the nuclear genome and 56.5% in the mitochondrial genome between species living in low (on average 0.357 Bq/g of dry sediment) and high radioactivity (on average 1.259 Bq/g of dry sediment). We also modeled the biologically effective dose of radioactivity received by each species (Received dose in μGy/h, *Figure 1*). This measure takes into account the transfer coefficient from environment to biota and the radio-toxicity of each radio-element for a crustacean model (ERICA tool V1.2.1 *Brown et al., 2016*). Again, we found positive correlations between the $d_S$/ra and the received dose of radioactivity (*Table 1*). These results are robust to the presence of influential cases and to variation in the model of evolution used to perform the test (see Methods, pGLS, p. values < 0.05).

As previously explained, the measured radioactivity at two sites overestimates the radioactivity level to which the organisms have been exposed for many generations because it is influenced by recent human activities. Moreover, while most species collected in highly radioactive habitats were from metamorphic or igneous formations, two species were from sedimentary formations (BOREON and GROTTAZE). Contrary to metamorphic and igneous formations, radioactivity in sedimentary formations is often observed in restricted localities (*Ielsch et al., 2017*) and can show large variations at the meter scale. A single radiation measurement may not therefore accurately represent the average radiation that a species was exposed to. To account for this variability as well as to include the two human-impacted sites into the regression analysis, we calculated the areal proportion of metamorphic and igneous rock within a 15 km radius around each site (later called λ15, *Figure 1*). This proportion was used as a proxy for the long-term regional radioactive exposure because the average linear distribution range of a groundwater crustacean is 30 km (*Eme et al., 2018*). We found a positive and stronger correlation between the $d_S$/ra and λ15 in both genomes (*Table 1*, *Figure 2*, n = 14 species). The linear model predicts that the nuclear and mitochondrial $d_S$/ra of a species living in a metamorphic formation (>50% of metamorphic and igneous rocks) are on average 34.4% and 61.3% higher, respectively, than those of a species living in a sedimentary formation (<50% of metamorphic or igneous rocks).

**Table 1.** Phylogenetic generalized least square (pGLS) regressions of the nuclear synonymous substitution rate ($d_S$/ra) and mitochondrial $d_S$/ra against the α radioactivity measured in the sediments (α radio.), the received dose (RD) of radioactivity, and the surface of metamorphic and igneous bedrock within a 15 km radius around the sampling sites (λ15).
α radioactivity and RD were log transformed to fit with the linear model assumptions. $R^2$ are Cox-Snell pseudo $R^2$.

| | Nuclear $d_S$/ra | | | | Mitochondrial $d_S$/ra | | | | |
| --- | --- | --- | --- | --- | --- | --- | --- | --- | --- |
| | Slope | L. ratio | p. value | $R^2$ | Slope | L. ratio | p. value | $R^2$ | N taxa |
| log(α radio.) | 0.034 | 5.995 | **0.014** | 0.393 | 0.506 | 7.895 | **0.005** | 0.482 | 12 |
| log(RD) | 0.038 | 6.51 | **0.011** | 0.419 | 0.491 | 5.981 | **0.015** | 0.392 | 12 |
| λ15 | 0.076 | 9.039 | **0.003** | 0.476 | 1.097 | 11.680 | **0.001** | 0.566 | 14 |

Each line corresponds to one likelihood ratio test between the models with and without the given explanatory variable. The impact of multiple predictors that are collinear are unreliable in the linear model framework (*Quinn and Keough, 2002*). As the three independent variables are collinear (R2 > 0.6) a model with a combination of these variable is not shown.

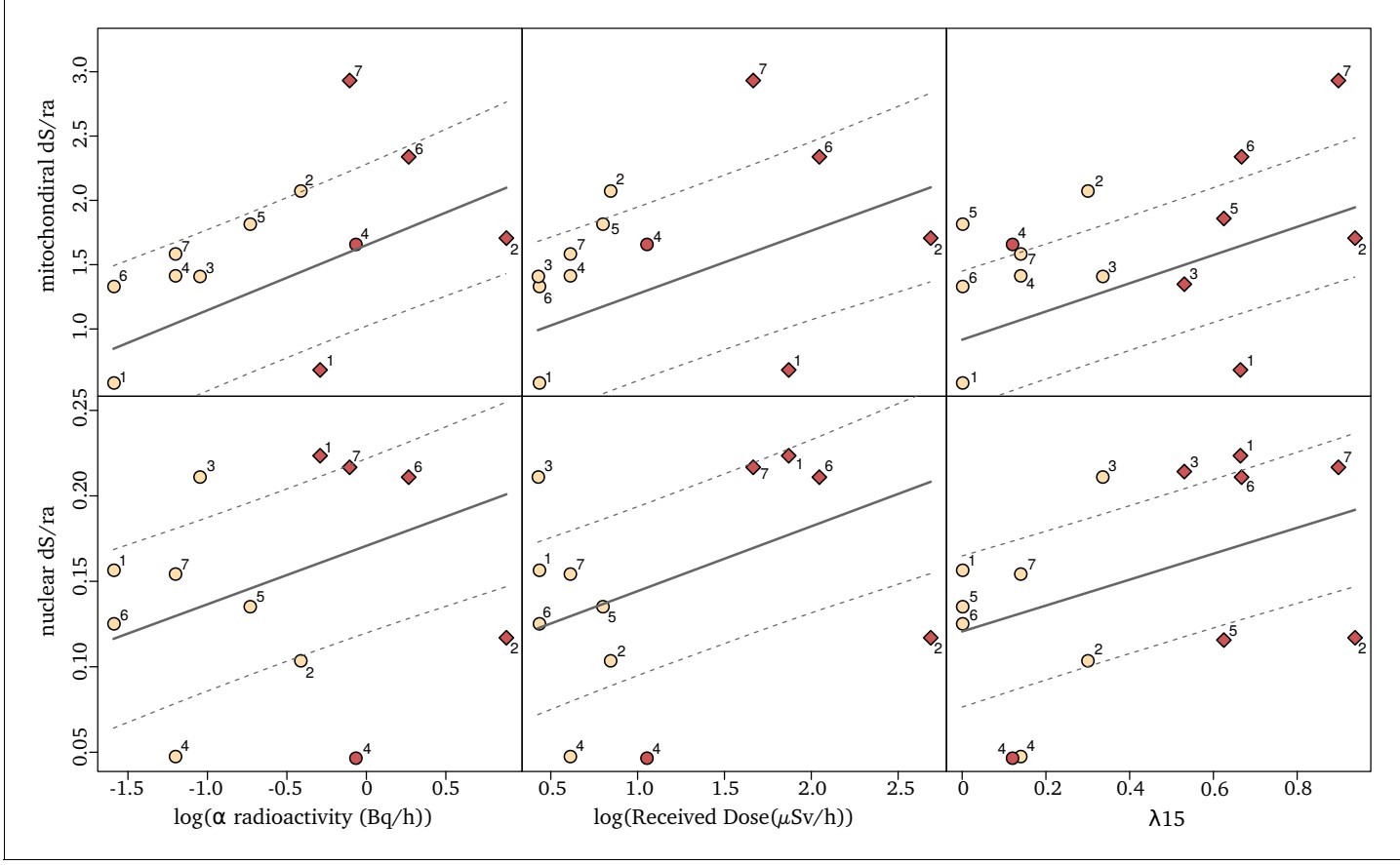

**Figure 2.** Relationships between synonymous substitution rate relative to the root age ($d_S$/ra) and radioactivity measured either as the sediment α radioactivity (left), the received dose (middle) or the proportion of igneous and metamorphic rock in a radius of 15 km around the sampling sites, λ15 (right). Each dot represents a species nuclear (top) and mitochondrial (bottom) $d_S$/ra. The fit of the pGLS model is indicated with a solid line and the confidence interval of the correlation is indicated with dashed lines. The two species of a pair are labeled with a number as in *Figure 1* (the number seven corresponds to the two species not forming a pair), with in red the species of the pair living in the highest α radioactivity. Species in sedimentary formations (λ15 < 50%) are depicted with circles and species in igneous/metamorphic formations (λ15 > 50%) with a diamond.

As bedrock radioactivity is positively correlated with the mutation rate, the underlying question is whether radioactivity also modifies the mutational spectrum, that is, the specific types of mutations that tend to occur. To address this question, we reconstructed the mutational spectrum of 6 independent pairs of species, each composed of two species located in low and high bedrock radiation set-ups, respectively, with a minimum of a 3X increase in the received dose of radioactivity between the two species (*Figure 1*). Briefly, we first estimated species polymorphism across a set of 2490 one-to-one orthologous genes by sequencing transcriptomes for eight individuals per species. After ancestral sequence reconstruction, we then identified mutations that occurred in each species and computed the relative proportion of each type of mutation (from A to T, A to C, ...), pooling together complementary mutations (e.g. p($C \rightarrow A$) + p($G \rightarrow T$) = p($C:G \rightarrow A:T$)). Two types of dependencies are present in testing mutational spectrum variation in a comparative data-set: (i) dependency among mutations – if the proportion of one mutation increases, the proportions of the other mutations will decrease – and (ii) the phylogenetic inertia, two closely related species have more chance to display more similar mutational spectrum. To take into account these two types of dependencies, we first used a forward selection approach as described in *Harris and Pritchard, 2017* to pull out mutations at different frequencies across habitats (see Materials and methods). Only the $C:G \rightarrow A:T$ mutation was significantly more frequent in radioactive habitats (*Table 2*, *Figure 3*). Second, we checked that these results were not induced by the phylogenetic structure of the data-set by using a pGLS regression between the proportion of each type of mutation and different radioactivity proxies. Again, we found a strong positive correlation between the proportion of $C:G \rightarrow A:T$

**Table 2.** Variation of the mutational spectrum as a function of bedrock radioactivity.

An ordered $\chi^2$ test is first used to test if mutation counts vary between low and high radioactive habitats while accounting for the dependency among mutation. In parallel, the phylogenetic dependency was taken into account using a Phylogenetic Generalized Least Square (pGLS) regression of the proportion of each mutation against the sediment $\alpha$ radioactivity ($\alpha$ radio.), Received Dose (RD), and areal proportion of metamorphic and igneous rock within a 15 km radius ($\lambda$15). $\alpha$ radioactivity and RD are log transformed to fit with the linear model assumptions. $R^2$ are Cox-Snell pseudo $R^2$.

| | Ordered $\chi^2$ test | pGLS regression | | | | | |
|---|---|---|---|---|---|---|---|
| Mutation type | Ordered p. value | Radio. | Slope | L.ratio | P.value | $R^2$ | N |
| P(C:G→A:T) | 0.000 | log($\alpha$ radio) | 0.013 | 13.010 | **0.000** | 0.662 | 10 |
| | | log(RD) | 0.014 | 12.079 | **0.006** | 0.635 | 10 |
| | | $\lambda$15 | 0.042 | 13.791 | **0.000** | 0.683 | 12 |
| P(A:T→T:A) | 0.111 | log($\alpha$ radio) | −0.009 | 6.819 | 0.072 | 0.433 | 10 |
| | | log(RD) | −0.011 | 9.348 | **0.030** | 0.541 | 10 |
| | | $\lambda$15 | −0.025 | 6.994 | 0.080 | 0.442 | 12 |
| P(C:G→T:A) | 0.351 | log($\alpha$ radio) | 0.012 | 3.323 | 0.612 | 0.242 | 10 |
| | | log(RD) | 0.010 | 1.684 | 0.776 | 0.131 | 10 |
| | | $\lambda$15 | −0.009 | 0.183 | 1.000 | 0.015 | 12 |
| P(A:T→C:G) | 0.520 | log($\alpha$ radio) | 0.001 | 0.121 | 1.000 | 0.010 | 10 |
| | | log(RD) | 0.000 | 0.015 | 1.000 | 0.001 | 10 |
| | | $\lambda$15 | 0.001 | 0.005 | 1.000 | 0.000 | 12 |
| P(C:G→G:C) | 0.982 | log($\alpha$ radio) | 0.005 | 2.137 | 0.864 | 0.163 | 10 |
| | | log(RD) | 0.006 | 2.403 | 0.726 | 0.181 | 10 |
| | | $\lambda$15 | 0.006 | 0.360 | 1.000 | 0.030 | 12 |
| P(A:T→G:C) | 0.982 | log($\alpha$ radio) | −0.022 | 7.778 | 0.075 | 0.477 | 10 |
| | | log(RD) | −0.019 | 4.004 | 0.180 | 0.284 | 10 |
| | | $\lambda$15 | −0.014 | 0.394 | 1.000 | 0.032 | 12 |

Each line of the pGLS tests corresponds to one likelihood ratio test between the models with and without the given explanatory variable. P values have been corrected following Holm's method (k=18).

mutations and bedrock radioactivity (**Table 2**, **Figure 3**). Selection is unlikely to be responsible for the increase of $C{:}G \to A{:}T$ as this correlation is also observed when the data set is limited to mutations found at the third, usually redundant, codon position (**Figure 3—figure supplement 1**, **Supplementary file 2**). A weak negative trend is observed for $A{:}T \to T{:}A$ mutations; however, this trend disappears when measured on third codon position.

Variations in the frequency of genetic variants among populations can originate from variation in the mutagenic environment or from biases in the fixation of mutations that can be due to demographic factors (**Mathieson and Reich, 2017**), natural selection (**Boussau et al., 2008**), or biased gene conversion (**Duret and Galtier, 2009**). While the impact of fixation biases on sequence evolution have been relatively well described, the long term impact of the mutagenic environment is less well-known. The most extensive demonstration of environmentally induced changes in the mutation spectrum comes from the study of carcinogens (**Seo et al., 2000**). Many carcinogens induce a specific mutational signature, for instance, UVs increase the $CC \to TT$ mutations whereas estrogen treatments increase the frequency of $A{:}T \to G{:}C$ mutations (**Flibotte et al., 2010**; **Nik-Zainal et al., 2015**). Here, we found an increase of the proportion of $G \to T$ mutations (and complementary $C \to A$) which is a mutation that increases in a variety of contexts and is not specific to a mutagen in particular. The occurrence of this mutation increases after exposure to carbon black (**Jacobsen et al., 2011**), to tobacco smoke (**Hollstein et al., 1991**; **Hainaut and Pfeifer, 2001**), or to polycyclic aromatic hydrocarbons (**Nik-Zainal et al., 2015**). Oxidative stress is the likely source of this mutation (**Wood et al., 1990**; **Besaratinia et al., 2004**; **Jacobsen et al., 2011**). Indeed, the characteristic damage linked to oxidative stress is the formation of 8-hydroxyguanine (**Shigenaga et al.,**

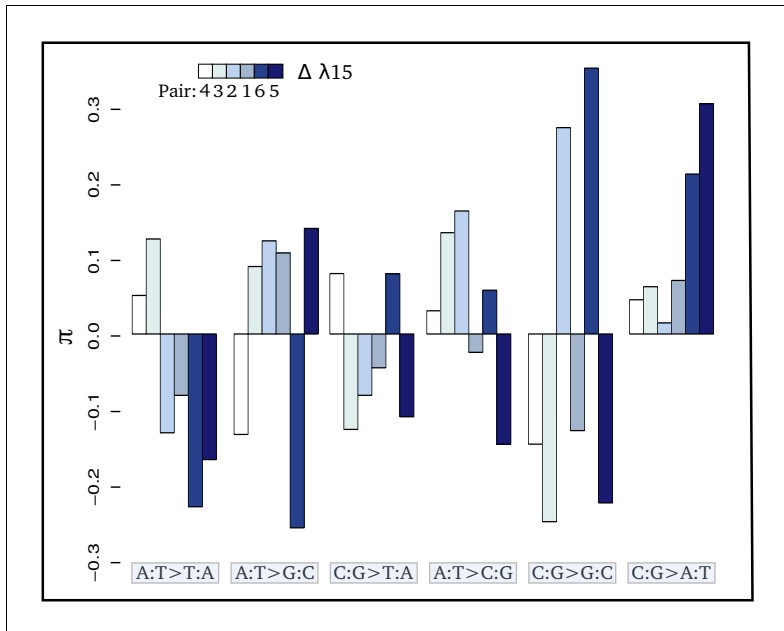

**Figure 3.** Contrasts (π) of the relative proportion of each mutation [p(i:j→k:l)] in each pair of sister species: $\pi_m = log \frac{p(i:j\rightarrow k:l)+}{p(i:j\rightarrow k:l)-}$ where + and - refer respectively to the species exposed to the higher and lower level of radioactivity in the pair m. Thus, positive bars represent a higher proportion of the given mutation in the species living in the high radioactivity rock. From left to right, bars are in increasing order of difference (Δ) in λ15 (the areal proportion of igneous and metamorphic rock in a radius of 15 km around the site) between the two species of each pair. From left to right, mutations are in increasing order of correlation with radioactivity. Numbers below the color scale indicate the species pair number as in *Figure 1*.

The online version of this article includes the following figure supplement(s) for figure 3:

**Figure supplement 1.** Contrasts (π) of the relative proportion of each mutation $[P(i:j\rightarrow k:l)]$ computed only on third positions in each pair of sister species: $\left[\pi_m = log \frac{P(i:j\rightarrow k:l)+}{P(i:j\rightarrow k:l)-}\right]$ where + and - refer respectively to the species exposed to the higher and lower level of radioactivity in the pair m.

1989) which leads to $G \rightarrow T$ mutations (*Shibutani et al., 1991*; *Cheng et al., 1992*). Thus, the mutational spectrum modification observed in radioactive environments is suggestive of an increase of the overall oxidative stress in these environments. Similarly, high artificial doses of ionizing radiation were found to increase oxidative damage (*Einor et al., 2016*; *Haghdoost et al., 2006*). As oxidative stress triggers more 8-hydroxyguanine formation in the mitochondrial genome than in the nuclear genome (*Richter et al., 1988*), this could explain the twice as strong effect observed in this study in the mitochondrial compartment.

Radioactive environments can cause mutations in two intertwined ways. First, the ionisation of molecules in the cells can directly affect the DNA structure by breaking the sugar phosphate backbone or can affect DNA indirectly through the radiolysis of water which decomposes the $H_2O$ molecules and create free radicals (*Desouky et al., 2015*). These free radicals can damage DNA molecules and create mutations. Second, radioactive decay chains also generate heavy metals (lead, polonium, etc) which are toxic for cells and also cause oxidative stress (*Quinlan et al., 1988*; *Pinto et al., 2003*). Due to the physicochemical association between radioactivity and heavy metals, in naturae correlative approaches alone cannot discriminate between the toxicity of heavy metals and the direct toxicity of radioactive rays.

While metamorphic and igneous rocks are more naturally radioactive than sedimentary rocks (*Ielsch et al., 2017*), they also display characteristic compositional and structural features which may also control species life history traits (*Cornu et al., 2013*; *Eme et al., 2015*). Here, we consider whether these characteristic features – specifically their poor calcium content and reduced habitat size and permeability – may confound the effect of radioactivity on the mutation rate. Metamorphic and igneous rocks are calcium poor relative to sedimentary rocks. In isopods, the cuticle is made of

calcium carbonate which is acquired by isopods from their environments (*Greenaway, 1985*). A lower calcium availability may slow down the growth of isopod species. However, a lower growth rate, if it slows down the generation time, is expected to decrease the mutation rate, while we observed the opposite. Because metamorphic and igneous rocks have smaller size and less permeable habitats, they could support smaller effective isopod population sizes. The $d_S$ is an estimator of the mutation rate that is not directly impacted by the effective population size (*Kimura, 1983*). However, as effective population size modulates the efficacy of natural selection, *Lynch, 2010* proposed that species with small effective population size should evolve higher mutation rate as a result of the accumulation of weakly deleterious mutations in genes involved in repair efficiency and replication fidelity. Efficacy of selection can be estimated by computing a transcriptome-wide ratio $d_N/d_S$. Here, we found no correlation between radioactivity and $d_N/d_S$ (pGLS, p.value = 0.3069) suggesting this effect is not at work in this data-set. Altogether, the influence of confounding factors that would drive the observed mutational variations is unlikely.

If oxidative stress is causing the increase of mutation rate, the radiolysis of water alone hardly explains the much higher impact of radioactivity on the mitochondrial mutation rate compared to the nuclear rate. As there is no reason to argue that radiolysis would not evenly occur within cells, it should impact both genomes similarly. However, differences between the two genomes may explain why the mitochondrial genome is more sensitive to radioactivity. First, the mitochondrial genome lacks some repair systems. For example, in a different yet analogous context, UV damage accumulates in the mitochondrial genome while they are repaired in the nuclear genome (*Clayton et al., 1974*). Second, the two genomes have very different organizations: while the nuclear genome is compacted into chromatin, the mitochondrial one is organized into nucleoids (*Chen and Butow, 2005*). While the role of the mitochondrial nucleoids is unclear, the chromatin structure and histone proteins protect the nuclear genome against radiation-induced strand breaks (*Ljungman, 1991*) and oxidative damage (*Ljungman and Hanawalt, 1992*). Third, direct radioactivity damage to the mitochondria may increase the activity of the mitochondrial respiratory chain and indirectly increase the production of ROS and DNA damage (*Yamamori et al., 2012*; *Kam and Banati, 2013*).

While life history traits are known to be central in controlling the mutation rate in most taxa (*Nabholz et al., 2008*; *Bromham et al., 1996*; *Nikolaev et al., 2007*; *Smith and Donoghue, 2008*), our results suggest that natural variation of radioactivity can have a comparable effect. Indeed, we found a minimum increase of around 30% percent of the nuclear mutation rate (60% in mitochondria) for species of waterlice living in the more naturally radioactive habitats made of igneous and metamorphic rocks. This increase is of the same magnitude as that observed when waterlice species evolve a 5-fold increase in generation time, a key life history trait controlling mutation rate in waterlice (*Saclier et al., 2018*) and organisms in general (*Thomas et al., 2010*; *Weller and Wu, 2015*; *Nabholz et al., 2008*). However, the influence of life history traits on the mutation rate varies widely among groups (*Allio et al., 2017*). Thus, the relative influence of radioactivity compared to life history traits could be different in groups like vertebrates, potentialy because of different gametogenesis (*Saclier et al., 2018*). Moreover, as groundwater waterlice ingest sediments (*Francois et al., 2016*), they are internally exposed to radioactivity, which may cause more mutations than through external exposure only (*Sawada, 2007*). The influence of environmental radioactivity on the mutation rate should therefore be explored across a wider range of organisms with contrasted diets and biologies.

Although the literature on the effect of low doses of radiation is far from being conclusive it often suggests a negligible biological impact (*Tubiana et al., 2006*; *Tubiana et al., 2009*) and only a handful of isolated studies support an impact of natural radioactivity on the mutation rate. For instance, a higher mutation rate was observed in the human mitochondrial genome in the Kerala region (*Forster et al., 2002*) and in satellite sequences of crickets inhabiting cave with high radon concentration (*Allegrucci et al., 2015*). In this study, by combining a large number of genes with the characteristics of the subterranean waterlice, namely the absence of UV confounding effect and limited dispersal, within a statistically powerful comparative framework allowing to work on large time scales and with numerous replicates, we found that a mild variation ($\simeq$ 3.5X) in natural bedrock radioactivity substantially alters the mutation rate, in particular the mitochondrial one. One key aspect that remains to be described is the shape of the relationship between the dose of radioactivity and the mutation rate: this study invalidates a model where low doses have no impact but falls short in differentiating between, for example, a linear and a hypersentivity (U-shaped) dependency model.

Altogether, while the universality of this finding warrants corroborative studies in other taxa, it suggests that the influence of natural radioactivity on the evolution of biodiversity has been overlooked.

## Materials and methods

### Sampling

To test the impact of radioactivity on molecular evolution, we focused on subterranean species belonging to the Asellidae family. Subterranean species are never exposed to UV radiation, live in contrasted bedrock set-ups and have very limited dispersal capacity (*Eme et al., 2018*), allowing us to make the assumption that different species have persisted in different but nearly constant radioactive habitats for numerous generations (but see Statistical analyses paragraph). One of the most interesting feature of subterranean Asellidae is their similarities in terms of morphology, lifestyle and life history traits. This high uniformity is likely due to a low rate of phenotypic evolution and a high level of convergence imposed by the subterranean lifestyle. As a result, distinguishing different species requires a high level of expertise and some species cannot be distinguished without molecular tools (*Morvan et al., 2013*). The birth of the Asellidae family is estimated at −350 My (*Morvan et al., 2013*). These characteristics allow us to compare species that are divergent enough to compute an accurate rate of molecular evolution but which conserved with very similar traits.

For 58 sites in France selected on the map of uranium (*Ielsch et al., 2017*), we collected Asellidae species and sampled about 50 g of sediment to measure global α radioactivity (see the following paragraph). Animals and sediments were collected using the Bou-Rouch pumping methods (*Bou and Rouch, 1967*). Collected specimens were stored in 96% ethanol at −20°C and were morphologically and molecularly identified. For molecular identification, DNA was extracted using an optimized chloroform DNA extraction protocol for the Aselloidea (*Calvignac et al., 2011*). We amplified DNA with primers targeting the 16S mitochondrial rDNA gene. PCR reactions were done following *Morvan et al., 2013*. PCR products were sequenced in both directions using the same primers as for amplification (GATC Biotech, Konstanz; Eurofins MWG Operon, Ebersberg; SeqLab, Göttingen, Germany; BIOFIDAL, Vaulx-en-Velin, France). Chromatograms were visualized and cleaned using Finch v1.5.0 (Geospiza, Seattle, USA). 16S have been deposited on the European Nucleotide Archive and are available under the accession number from LR214526 to LR214880. Using *Eme et al., 2018* molecular species delimitation, each sequence has been assigned to a species. Based on this taxonomic assignment and radioactivity measurement, 14 species were retained for further analyses (*Supplementary file 1*). For these 14 selected species, during a new sampling trip, individuals were flash frozen alive in the field.

### Measures of radioactivity

#### α radioactivity

In order to estimate the global radioactivity in sediments, we measured the α radioactivity. An α decay occurs when an atom disintegrates by ejecting an α particle, that is, a particle made of two neutrons and two protons. The α radioactivity should be correlated with the global radioactivity in natural systems. For the 58 prospected sites, three samples of about 50 g of sediment were collected in polyethylene bottles. α radioactivity measurements were made by the LABRADOR service (Institut de Physique Nucléaire de Lyon, France) on proportional counter with the NF ISO 18589–6 standard (Data available on Zenodo, DOI: 10.5281/zenodo.4071754).

#### Received dose

In order to estimate the received dose of radiation that is impacting organisms, we collected three samples of 100 g of fine sediments (<100 μm) in each of the selected sites. These sediments were prepared with the NF EN ISO 18589–2 standard and measured by gamma spectrometry in conformity with the NF EN ISO 18589–3 standard using the PRISNA-P analysis platform at the Centre d'Etude Nucléaire de Bordeaux Gradignan (CENBG). This platform is certified by the French Nuclear Security Authority (ASN) for measures of natural radioactivity. Samples were dried in open air, and then dried at 100°C. Matters were packed in a waterproof geometry. Geometries were sealed for one month and then counted for a duration of 86500 s on the same chain of measure. The chain used is an ORTEC chain, presenting an efficiency of about 60% and calibrated in May 2016. This

chain is equipped with a cosmic veto device and located in a half buried laboratory in order to: (i) attenuate the background noise, (ii) improve the detection limits, and (iii) reduce the measure uncertainty. The activity of the main radionuclides were measured in sediment and the activity of the remaining radionuclides was deduced based on the hypothesis of a secular equilibrium of the uranium 238 and thorium 232 chains. As activities of the radionuclides of the uranium 235 decay chain are generally low, only measures higher than the decision threshold (according to the measure variability) were taken into account. When the uranium 235 activity was too low to be measured it has been deduced from the uranium 238 activity, using the natural isotopic ratio of 21.6.

The received dose impacting organisms was estimated using the ERICA tool (V1.2.1, *Brown et al., 2016*) with a 'crustacean' model. We assumed that organisms stay 10% of their time on the surface of sediment and 90% inside sediment. All radionuclides available in the tool were taken into account (i.e, $U^{238}$, $Th^{234}$, $U^{234}$, $Th^{230}$, $Ra^{226}$, $Pb^{210}$, $Po^{210}$, $U^{235}$, $Th^{231}$, $Pa^{231}$, $Th^{227}$, $Th^{232}$, $Ac^{228}$ and $Th^{228}$). We used the distribution coefficients proposed by the ERICA tool. Concentration factors proposed by the tool were used when available. If not, we used the concentration factor of the closest biogeochemical element available.

Two sites (BRETEMIN and BOREON) show a disruption of the secular equilibrium in the $U^{238}$ chain. This suggests that nearby industrial activities (e.g. lead mines) have modified the natural radioactivity of these two sites. As these industrial activities are very recent (since 1950), their impact on the substitution rate which is measured on a much longer time scale is unlikely. These two sites were removed from the correlation between $d_S$/ra and radioactivity measured with the global $\alpha$ radioactivity or with the received dose.

## Proportion of magmatic and igneous rocks in a 15 km radius

Using the geological map of France (scale : 1/1,000,000, BRGM), the areal proportions of magmatic and igneous rocks in a radius of 15 km around sampling sites were computed (noted λ15), 30 km represents the average distribution range for a subterranean isopod (*Eme et al., 2018*).

## Transcriptome sequencing and assembly

### Sequencing

For each species, we sequenced transcriptomes from eight individuals. For each individual total RNA was isolated using TRI Reagent (Molecular Research Center). Extraction quality was checked on a BioAnalyser RNA chip (Agilent Technologies) and RNA concentrations were estimated using a Qubit fluorometer (Life Technologies). Prior to any additional analysis, species identification was corroborated for each individual by sequencing a fragment of the 16S gene. Illumina libraries were then prepared using the TruSeq RNA Sample Prep Kit v2 (Illumina). For each species one library was paired-end sequenced using 100 cycles, and the seven other libraries were single-end sequenced using 50 cycles on a HiSeq2500 sequencer (Illumina) at the IGBMC GenomEast Platform (Illkirch, France). We obtained around 30 million single-end reads per individual and 118 million paired-end reads per species.

### Assembly

Adapters were clipped from the sequences, low quality read ends were trimmed (phred score <30) and low quality reads were discarded (mean phred score <25 or if remaining length <19 bp) using fastq-mcf of the ea-utils package (*Aronesty, 2013*). Paired-end transcriptomes were de novo assembled using Trinity v2.3.2 (*Grabherr et al., 2011*). Open reading frames (ORFs) were identified with TransDecoder (http://transdecoder.sourceforge.net). For each assembled component, only the most express ORF was retained.

## Families of orthologous genes

Gene families were delimited using an all-against-all BLASTP (*Altschul et al., 1990*) and SiLix (*Miele et al., 2011*) on the ORFs delimited in the previous step. We then kept gene families containing the 14 species, with only one sequence for each species in order to remove paralogs. We obtained 2490 families hereafter considered as one-to-one orthologous genes. These genes were aligned with PRANK (*Löytynoja and Goldman, 2008*) using a codon model and sites ambiguously aligned were removed with Gblocks (*Castresana, 2000*).

## Species tree and gene trees

The 2490 genes were concatenated and a phylogenetic tree (hereafter called the concatenated tree) was built using PhyML v3.0 (*Guindon et al., 2010*) under a GTR+G+I model with 100 bootstrap replicates and was rooted using the Slavus lineage (*Proasellus boui* and *Proasellus slavus*) as an outgroup (*Morvan et al., 2013*). Most nodes have a bootstrap value of 100% (*Figure 1—figure supplement 1*). Two nodes have values at 84% and 98% in the clade containing P. nsp VIELVIC; P. nsp HYPOPRAT; P. nsp MONTBAR and P. nsp ROSSFELD. To check the relationship between these four species, we built 2490 individual gene trees with PhyML v3.0 under a GTR+G+I model with 100 bootstrap replicates. Twenty-nine gene trees strongly support (bootstraps >90%) the phylogeny of the concatenated tree for this clade, 208 support other various topologies and the remaining 2253 gene trees do not support any relationship in particular for this clade (bootstraps <90%). Thus, the phylogeny for this clade remained unresolved, possibly as the consequence of a concomitant speciation process of these four species. For approaches with pairs of sister species, as we were unable to resolve the phylogeny for this clade, we selected the species living in the highest level of radioactivity (*P. nsp* HYPOPRAT) and the species living in the lowest level of radioactivity (*P. nsp* MONTBAR) among these four species to build a pair, resulting in a total of 6 pairs of sister species (sensu *Felsenstein, 2004*).

## Mitochondrial genes

Mitochondrial genes were not present amongst the 2490 genes obtained above. Indeed, owing to a different genetic code in invertebrate mitochondria, mitochondrial ORFs were systematically missed by the ORF caller (Transdecoder). We reconstructed mitochondrial genomes using the de novo transcriptome assemblies. Large mitochondrial contigs were built with MITObim (*Hahn et al., 2013*) by using RNA-seq reads. These contigs were mapped on the assembled mitochondrial genome from the closest possible species (taken from *Saclier et al., 2018*), allowing us to assemble them. Mitochondrial genomes were annotated using the MITOS web server (*Bernt et al., 2013*). We recovered the 13 mitochondrial protein-coding genes. Mitochondrial genes were aligned with PRANK (*Löytynoja and Goldman, 2008*) and sites ambiguously aligned were removed with Gblocks (*Castresana, 2000*).

## Rate of molecular evolution

We used the synonymous substitution rate ($d_S$) computed on the terminal branches of the tree as a proxy for the long-term species mutation rate (*Kimura, 1983*). This proxy is valid in absence of selection on codon usage. To check for the absence of biased codon usage, we computed the effective number of codons on the 2490 orthologous genes (ENC, *Wright, 1990*). This number varies between 20 (only a single codon is used for each amino acid) and 61 (all synonymous codons are used with equal frequency for each amino-acid). ENC ranged between 49.17 and 50.48 (*Supplementary file 1*), indicating a moderate codon usage bias, more importantly, they do not correlate with alpha radioactivity (pGLS, p.value = 0.6378). Altogether, the $d_S$ estimation does not seem impacted by a strongly biased or variable codon usage.

To compute $d_S$ we first removed some genes showing a conflicting phylogeny. Including genes supporting different phylogenies in a concatenation amounts to constrain a wrong phylogeny for these genes which may biases $d_S$ estimations. Indeed imposing a wrong gene tree will tend to generate convergent mutations in terminal branches of the tree. To avoid such bias in our $d_S$ estimation we used ProfileNJ (*Noutahi et al., 2016*) with a bootstrap threshold of 90% to compute a cost of reconciliation between the concatenated tree and the gene trees. We kept the gene families with a cost of reconciliation of zero and with sequences long enough for all species (at least a half of the alignment) and removed all other genes, resulting in a set of 769 gene families. $d_S$ were estimated using CoEvol (*Lartillot and Poujol, 2011*). This software program implements a Muse and Gaut codon model (*Muse and Gaut, 1994*), with Brownian variation in $d_S$ and $d_N/d_S$ along the tree. Bayesian inference and reconstruction of the history of variation in $d_S$ and $d_N/d_S$ along the tree is conducted by Markov Chain Monte Carlo (MCMC). Two independent chains were run, and were stopped after checking for convergence by eye and with the tracecomp program included in the Coevol package (effective sample size >200 and discrepancy between chains < 0.3). Chains were stopped after 7117 generations (4200 generations excluded as burn-in). The age of the root was

arbitrarily set to 1, resulting in synonymous substitution rate estimates that are relative to the root age ($d_S$/ra) (**Supplementary file 1**). In order to ensure that assumptions made by CoEvol on the $d_S$ evolution along branches don't bias the $d_S$/ra estimation, $d_S$ were also computed with CodeML (**Yang, 2007**) using a free ratio model and with the Bio++ suite (**Dutheil and Boussau, 2008**). For the last one, a non-homogeneous model (NY98 model) was first applied to the alignment with BppML and then the MapNH program (Version 1.1.1) of the TestNH package (**Guéguen and Duret, 2018**) was used to reconstruct the ancestral states to estimate the number of synonymous substitutions on each branch. We multiplied the CoEvol $d_S$/ra by the time estimated, to obtain a classical $d_S$. This CoEvol $d_S$ was highly correlated with CodeML $d_S$ ($R^2 = 0.81$) as well as with Mapnh $d_S$ ($R^2 = 0.82$). Regarding the correlation with radioactivity, by dividing the CodeML $d_S$ and the Mapnh $d_S$ by the divergence time estimated by CoEvol in order to obtained comparable $d_S$/ra among all species, we obtained similar results whatever the method used to compute the $d_S$/ra (**Supplementary file 3**).

## Mutational spectrum

To compute the mutational spectrum, we used an approach by pairs of sister species. We determined the polymorphism at the population level for each species by mapping the seven single-end transcriptomes on the assembled paired-end transcriptome with BWA (**Li and Durbin, 2009**). BAM files were produced with SAMtools (**Li et al., 2009**), and reads2snps (**Gayral et al., 2013**) was used to detect polymorphic sites. We then conserved only the 2490 orthologous genes shared by all species to compute the mutational spectrum on the same set of genes.

For the two species of a pair, we reconstructed the ancestral sequence using a parsimonious approach. Namely, for each site in the alignment, if the two species had a single shared allele, this allele was considered as ancestral and the other alleles, if they existed, were considered as derived from the ancestral allele. For each species, we estimated the probability of a mutation in their population, $p(i \rightarrow j | f(i)_{anc})$, by counting each type of mutation, either on all positions or on third positions, corrected by the ancestral base frequency:

$$p(i \rightarrow j | f(i)_{anc}) = \frac{N(i \rightarrow j)}{Ni_{anc}}$$

This probability being dependent on the mutation rate μ, we estimated the mutational spectrum by the proportion, when a mutation occurs, of mutation from the base i to the base j, noted $p(i \rightarrow j | \mu, f(i)_{anc})$:

$$p(i \rightarrow j | \mu, f(i)_{anc}) = p(i \rightarrow j | f(i)_{anc}) * \frac{1}{\sum_{i=\{A,C,G,T\}} \sum_{j=\{A,T,C,G\}} p(i \rightarrow j | f(i)_{anc})}$$

This proportion takes into account the mutation rate and is so comparable across species. We pooled complementary mutations (e.g. A to C with T to G) to increase the counts by mutational categories and improve statistical power.

## Statistical analyses

Correlations between $d_S$/ra computed on terminal branch of the tree and the different measures of radioactivity were tested using phylogenetic Generalized Least Squares models (pGLS **Martins and Hansen, 1997**) with the nlme (**Pinheiro et al., 2007**) and ape packages (**Paradis et al., 2004**) in R (**R Development Core Team, 2020**). As $d_S$/ra is computed on terminal branches of the tree, this test assumes that radioactivity remained stable along the time represented by this branch which is disputable. Natural radioactivity has been slowly decreasing for the last 2 Gy (**Karam and Leslie, 2005**). However, this decrease is global and we can thus consider that the delta of radioactivity is stable over time. Second, the terminal branches on which the $d_S$/ra is estimated do not exceed 10 My, a time frame in which the level of radioactivity can be considered stable. Thus, the present quantitative estimates should be representative of the radioactivity contrasts that persisted between these habitats in the time-frame of this study. That said, even if subterranean species have low dispersal abilities, there distribution have changed over time, but these movements could blur a signal and are unlikely to generate one.

For the α radioactivity and received dose, the two species showing a disruption in the secular equilibrium were removed. The ultrametric tree built by CoEvol was used to calculate the

phylogenetic variance/covariance matrix under a Brownian motion model to take into account the non-independence among species in the pGLS. Normality of residuals was checked for all models, log transformation was applied when the normality was rejected (Shapiro test).

For mutational spectrum, because the proportions of each mutation are not independent from each other, we proceeded to a selection of variables as proposed by *Harris and Pritchard, 2017*. This procedure consists of performing iterative chi-square tests. To achieve that, for each mutation we summed the counts of the six species living in 'highly' radioactive environments that we compared to the counts of the six species living in 'weakly' radioactive environments. First, 'normal' chi-square tests are performed for each mutation. Mutations are then ordered following p.values of these tests that we call unordered(p). An ordered p.value is then computed. For the mutation with the lowest unordered(p):m1 the p.value remains unchanged (ordered(p)=unordered(p)). For the following mutation with the second lowest unordered(p): m2, an ordered(p) is computed by doing a chi-square test between the count of this mutation (m2) and the sum of the counts of the other mutations excepted the first one (sum of m3 to m6), and so one for each mutation. For the last mutation (m6), ordered(p) is computed by doing a chi-square test between m5 and m6 counts. To take into account the phylogenetic inertia, we then tested the correlation between the proportion of each mutation and radioactivity ($\alpha$ radioactivity, received dose of radioactivity or proportion of metamorphic or igneous rocks) with a pGLS, this time pruning P. nsp VIELVIC and P. nsp MONTBAR from the chronogram built by CoEvol. P.values from pGLS were corrected with Holm's method to adjust for the multiple tests and to control the false discovery rate.

For each tests, pGLS assumptions, namely normality of residuals, homoscedasticity, absence of influential cases, an evolutionary process (here a brownian motion), have been checked. Normality was checked by plotting residuals of models. Homoscedasticity was checked by plotting the residuals against the fitted values of models. Absence of influential cases was tested by a jackknife approach consisting in removing one by one each species of the data and redoing the test. Robustness to the evolutionary model was tested by performing the pGLS with the four main evolutionary models (Blomberg, Martins, Pagel and Brownian).

## Acknowledgements

This work was supported by the French program STYGOMICS (CNRS Défi Enviromics), the Zone Atelier Territoire Uranifère, the Grottes d'Azé, the Conseil Départemental de Saône-et-Loire, the Association Culturelle du Site d'Azé and the Agence Nationale de la Recherche (ANR-15-CE32-0005 Convergenomix, France, ANR-17-EURE-0018 H2O'Lyon, France). We gratefully acknowledge support from the CNRS/IN2P3 Computing Center (Lyon/Villeurbanne, France) for providing a significant amount of the computing resources needed for this work and the CERESE (Univ. Lyon 1) for the storage of the biological material. We thank the Grottes d'Azé and more specifically Lionel Barriquand, the Mercantour national parc (authorization n°2015-251) and more specifically its scientific manager Marie-France Leccia, the Fédération Rhône-Alpes de Protection de la Nature and the owner of Bout du monde mine for giving us access for sampling. We thank Marcel Meysonnier, Aymeric and Audric Berjoan, Josiane and Bernard Lips, Audrey Brechet, Claude Bou, Benjamin Benti and Léa Dantony for their help in the field. We are grateful to Nicolas Lartillot, Gilles Escarguel, Marie Sémon, Laurent Guéguen, Nicolas Galtier, Benoît Nabholz and Bastien Boussau for helpful discussions. We also thank Laurent Simon and Laura Grice for their suggestions in the latter stages of manuscript preparation.

## Additional information

### Funding

| Funder | Grant reference number | Author |
| --- | --- | --- |
| Centre National de la Recherche Scientifique | STYGOMICS - Défi enviromix | Patrick Chardon<br>Florian Malard<br>Lara Konecny-Dupré<br>Tristan Lefebure<br>Christophe J Douady |

| Agence Nationale de la Recherche | ANR- 15-CE32-0005 Convergenomix | Lara Konecny-Dupré Laurent Duret Tristan Lefebure Christophe J Douady |
|---|---|---|
| H2O'Lyon, France | ANR-17-EURE-0018 | Christophe J Douady |

The funders had no role in study design, data collection and interpretation, or the decision to submit the work for publication.

## Author contributions

Nathanaëlle Saclier, Conceptualization, Data curation, Formal analysis, Investigation, Methodology, Writing - original draft, Project administration, Did the field work, defined orthologous gene families, built the phylogeny, computed substitution rate, performed statistical analyses and computed the mutational spectrum; Patrick Chardon, Conceptualization, Data curation, Formal analysis, Funding acquisition, Methodology, Writing - review and editing, Did the field work, performed all radioactivity measurements and the received dose estimation; Florian Malard, Conceptualization, Data curation, Formal analysis, Supervision, Funding acquisition, Validation, Investigation, Methodology, Project administration, Writing - review and editing, Did the field work, dissected and identified morphologically all sampled species; Lara Konecny-Dupré, Data curation, Formal analysis, Supervision, Writing - review and editing, Extracted DNA and RNA from all samples, made PCR and migrations and prepared library for sequencing; David Eme, Data curation, Validation, Investigation, Writing - review and editing, Did the field work, dissected and identified morphologically all sampled species; Arnaud Bellec, Formal analysis, Writing - review and editing, Extracted the outcrop cover of low-radioactivity sedimentary rocks and high-radioactivity metamorphic and igneous rocks in a radius of 15 km around the sampling, and made the map of uranium for Figure 1; Vincent Breton, Validation, Writing - review and editing; Laurent Duret, Validation, Methodology, Writing - review and editing, Supervised the computation of the mutational spectrum; Tristan Lefebure, Conceptualization, Data curation, Formal analysis, Supervision, Funding acquisition, Investigation, Methodology, Writing - original draft, Project administration, Writing - review and editing, Did the field work, assembled transcriptomes and defined ORFs and orthologous gene families; Christophe J Douady, Conceptualization, Data curation, Formal analysis, Supervision, Funding acquisition, Validation, Investigation, Methodology, Writing - original draft, Project administration, Writing - review and editing, Did the field work

## Author ORCIDs

Nathanaëlle Saclier https://orcid.org/0000-0003-1522-9644
David Eme https://orcid.org/0000-0001-8790-0412
Laurent Duret http://orcid.org/0000-0003-2836-3463
Tristan Lefebure https://orcid.org/0000-0003-3923-8166
Christophe J Douady https://orcid.org/0000-0002-4503-8040

## Decision letter and Author response

Decision letter https://doi.org/10.7554/eLife.56830.sa1
Author response https://doi.org/10.7554/eLife.56830.sa2

# Additional files

## Supplementary files

• Supplementary file 1. Fourteen sequenced species with sampling coordinates, synonymous substitution rate relative to the root age for nuclear and mitochondrial genome, $\alpha$ radioactivity measured on each site, effective dose of radioactivity (in $\mu Gy/h$), and this effective dose corrected for recent human impact (in $\mu Gy/h$), non-synonymous substitution rate over synonymous substitution rate ($d_N/d_S$), the areal proportion of magmatic and metamorphic rocks in a radius of 15 km around the sampling point ($\lambda 15$), the GC content for all positions or for third positions, the effective number of codon (ENC) and the raw $d_S$ computed with the CodeMl programm and with the MapNH programm (not relative to root age).

• Supplementary file 2. Statistical tests for the mutational spectrum analysis. An ordered Chi$^2$ test as described in *Harris and Pritchard, 2017* has been performed on mutation counts computed on third positions. A Phylogenetic Generalized Least Square (pGLS) regression of the proportion of each mutations against the α radioactivity measured in sediment (α radio.), the Received Dose (RD) modeled with ERICA tool, and the areal proportion of metamorphic and igneous rock within a 15 km radius (λ15) has also been performed. α radioactivity and RD were log transformed to fit with linear model assumptions. $R^2$ are Cox-Snell pseudo $R^2$.

• Supplementary file 3. Phylogenetic Least Square (PGLS) regression of $d_S$ computed with CoEvol, CodeMl, or mapNH against radioactivity measured as the α radioactivity measured in sediments, as the effective dose received by organisms or as the areal proportion of metamorphic and magmatic rocks in a radius of 15 km around the sampled point (λ15). Each test corresponds to one likelihood ratio test between the models with and without the given explanatory variable. For α radioactivity and received dose, sampled sites with a break in the secular equilibrium were removed, resulting in tests with only 12 taxa.

• Transparent reporting form

### Data availability

16S sequences have been deposited on the European Nucleotide Archive and are available under the accession numbers from LR214526 to LR214880 (https://www.ebi.ac.uk/ena/data/view/LR214526-LR214880). Alignments and the list of genes used to compute synonymous substitution rate have been deposited on Zenodo (https://zenodo.org/deposit/2563829). Transcriptome reads have been deposited on the European Nucleotide Archive and are available under accession numbers from LR536601 to LR536626 in the study ID PRJEB14193 (https://www.ebi.ac.uk/ena/data/search?query=PRJEB14193). Number of reads and data used for correlations, namely measures of radionuclides and mutations counts have been deposited on Zenodo (https://doi.org/10.5281/zenodo.4071754).

The following datasets were generated:

| Author(s) | Year | Dataset title | Dataset URL | Database and Identifier |
|---|---|---|---|---|
| Saclier N, Chardon P, Malard F, Konecny-Dupré L, Eme D, Bellec A, Breton V, Duret L, Lefébure T, Douady CJ | 2020 | Data used in the article | https://doi.org/10.5281/zenodo.4071754 | Zenodo, 10.5281/zenodo.4071754 |
| Saclier N, Chardon P, Malard F, Konecny-Dupré L, Eme D, Bellec A, Breton V, Duret L, Lefébure T, Douady CJ | 2020 | Bedrock radioactivity influences the rate and spectrum of mutation - Orthologous genes | https://doi.org/10.5281/zenodo.2563829 | Zenodo, 10.5281/zenodo.2563829 |
| Saclier N, Chardon P, Malard F, Konecny-Dupré L, Eme D, Bellec A, Breton V, Duret L, Lefébure T, Douady CJ | 2019 | Aselloidea isopods Sanger sequencing | https://www.ebi.ac.uk/ena/browser/view/LR214526-LR214880 | ENA, PRJEB30668 |
| Saclier N, Chardon P, Malard F, Konecny-Dupré L, Eme D, Bellec A, Breton V, Duret L, Lefébure T, Douady CJ | 2019 | Aselloidea isopods transcriptomes sequencing and denovo assembly | https://www.ebi.ac.uk/ena/data/search?query=PRJEB14193 | ENA, PRJEB14193 |
| Lefébure T, Saclier N | 2019 | 16S sequences | https://www.ebi.ac.uk/ena/data/view/LR214526- | EBI European Nucleotide Archive, |

| | | | LR214880 | LR214526-LR214880 |
|---|---|---|---|---|
| Lefébure T, Saclier N | 2019 | Transcriptome reads | https://www.ebi.ac.uk/ena/data/view/LR536601-LR536626 | EBI European Nucleotide Archive, LR536601-LR536626 |

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
