## [Decision Letter]

**Acceptance summary:**

We still know little about how variation in natural levels of radioactivity impact germline mutation rates. By contrasting substitution rates across 14 waterlice species, the study reveals an increase in mutation rates in more radioactive environments; in particular of G>T mutations, consistent with oxidative stress. Thus, this comparative approach identifies the impact of an external mutagen on de novo mutation rates.

**Decision letter after peer review:**

Thank you for submitting your article "Bedrock radioactivity influences the rate and spectrum of mutation" for consideration by *eLife*. Your article has been reviewed by three peer reviewers, one of whom is a member of our Board of Reviewing Editors, and the evaluation has been overseen by Detlef Weigel as the Senior Editor. The following individual involved in review of your submission has agreed to reveal their identity: Shamil R Sunyaev (Reviewer #2).

The reviewers have discussed the reviews with one another and the Reviewing Editor has drafted this decision to help you prepare a revised submission.

As you can see from the individual reviews below, the reviewers all appreciated the study design and found the results convincing and interesting. Nonetheless, they raised a number of concerns that will need to be addressed in revisions. The most serious set of concerns pertain to the interpretation of the findings. More discussion is needed of possible confounders to the association of bedrock radioactivity levels and substitution rates. Moreover, the assumption that the environmental conditions remain constant in a lineage seems tenuous, which casts doubt on the quantitative estimates provided. This point too merits further discussion. Finally, the interpretation in terms of ROS seems excessively strong. The reviewers also make a number of useful suggestions about statistical analyses, notably in terms of the analysis of the mutation spectrum. In terms of presentation, it was also felt that reviews of the field in the Introduction and Discussion could be improved substantially.

Reviewer #1:

I really liked the idea of comparing synonymous substitution rates and types in closely related species that experience different levels of bedrock radioactivity. I am convinced that there seems to be an association there, and the mutation spectrum results are also interesting, but it was hard to evaluate rival hypotheses for the association without knowing more about the species that live in environments with lower vs higher levels of radioactivity. I was also curious how diverged they are at the nucleotide level. Given that the authors have the data to look at these questions, it would be helpful to know how diversity levels and the allele frequency spectrum differs between different species compare for instance.

I also didn't quite know what to make of the quantitative estimates when I expect the level of bedrock activity likely changed quite a bit over the o(Ne) generations of the species. Could the authors comment on that?

For the mutation spectrum analysis, I think the authors should use a forward variable selection approach, such as the one employed in Harris and Pritchard, 2017.

In terms of presentation, I found the Introduction somewhat unhelpful, in that it lumps references for all sorts of ionizing radiation and UV, on both point mutations and microsatellites. For this reader at least, it would be helpful to overview what types have been studied, and what is known specifically for the type examined here. Also this study should be discussed: https://www.ncbi.nlm.nih.gov/pubmed/25809527. On a related but more minor note, I was unconvinced by the argument that this question cannot be studied experimentally and I don't think the argument is needed to motivate their approach.

In turn, the discussion of life history trait effects on mutation rates in metazoans was both too strong and oddly referenced. As examples, Martin and Palumbi, 1993 is actually an argument for consideration the effect of metabolic rates; Saclier et al., 2018 is only in isopods, when the claim is made for metazoans etc… Similar to my comment on the Introduction, I think a more systematic discussion of the literature is needed.

Finally, I thought the authors could do more to link their findings to studies in other organisms, in particular humans, where there are a number of studies of mutation patterns in populations living in diverse environments (e.g., https://journals.plos.org/plosgenetics/article?id=10.1371/journal.pgen.1006581).

Reviewer #2:

This manuscript is proposing an attractive model suggesting that natural radioactivity (even at a low level) affects the mutation rate in waterlice. The authors collected unique biological material in a number of areas with variable radioactivity levels. The data and the proposed model are certainly of great interest. However, I find that the manuscript would benefit from a few additional statistical analyses of the data and from softening the discussion about ROS.

1) The analysis makes a major assumption that the habitat and the corresponding level of radiation remains unchanged after speciation. This assumption warrants a discussion. Overall, I agree that a violation of this assumption would result in a conservative estimate and main results should not be affected.

2) I think that the study lacks a test for robustness of pGLS.

3) The manuscript should discuss potential covariates and confounders. Is it possible that variables other than radioactivity level are responsible for the signal?

4) The conclusions of the manuscript are based on Figure 2 showing non-ulotrametric trees and ratios of synonymous branch lengths. It would be great to show the data for "polymorphic" variants (just terminal branches).

5) In human cells, radiation primarily induces deletions associated with the microhomology repair. If this is also the case in the waterlice system, the authors should be able to find a highly specific and strong signal on top of the results about point mutations. This would dramatically strengthen the conclusions.

6) I do not find that the shift in mutation spectra is a sufficiently strong evidence in favor of ROS. This either requires a much stronger argument, or the discussion about the role of oxidative damage should be softened.

Reviewer #3:

In this study, Saclier et al. explore the effects of natural radioactivity on the germline mutation rate and spectra of subterranean waterlice, aiming to address the question of whether radioactive habitats have long-term evolutionary effects on the genomes of species which inhabit them. The use of this organism to test such a fundamental question in evolutionary biology is very clever, as waterlice are not exposed to confounding effects of UV radiation and have naturally limited dispersal. I found the paper well-written and a fun topic to read, and the arguments that natural radioactivity can have a tangible impact on genome evolution were compelling.

I have a few concerns and suggestions about the regression models and statistical analyses.

Figure 2: This would be easier to follow if points were labeled/colored/shaped to indicate the corresponding species, site, and bedrock formation type.

Table 1: The authors use independent regression models to test for the associations between mutation rate and each three variables of interest: α radioactivity, Received Dose, and λ-15. Why did the authors perform three separate simple regressions rather than a multiple regression using all three of these variables as covariates? Given that λ-15 and Received Dose appear to be stronger predictors of mutation rate than α radioactivity, does this mean the statistically significant association of α radioactivity goes away when adjusting for the other two variables? I expect these three variables to be highly collinear and too many covariates could easily result in overfitting with such a small sample, but the claims drawn from this table would benefit from a more nuanced statistical analysis.

Also, were there any additional environmental covariates that may have been measured when samples were collected or integrated from other data sources (e.g., latitude, distance to nearest industrial activities, etc.)? If possible, it would be interesting to see if other environmental factors can also explain the variation in mutation rates, but I recognize that such data may not be available, and even if it were, this analysis may not be feasible with a small sample.

Table 2: Performing 3 separate regressions for each of 6 mutation types results in some multiple testing issues that must be addressed, at a conservative Bonferroni-corrected α value of.05/18=.0028, only the C:G>A>T mutation class is statistically significant across all 3 explanatory variables, but the A:T>T:A class is not. This multiple testing burden could perhaps be alleviated by using multiple regression as suggested above. Further, the response variables of the 6 regressions are proportions that add to 1, so these are not independent statistical tests, even though they are analyzed and presented as such. Is there a different statistical model that can be used that takes into account the interdependence of the 6 components of the mutation spectrum?

How well-correlated are the nuclear and mitochondrial dS/ra values across samples? A priori, I expect them to be strongly correlated, but it would be interesting and straightforward to investigate if radioactivity levels also affected relative differences in nuclear and mitochondrial mutation rates, e.g., perhaps radioactivity increases nuclear mutation rates in a linear fashion, but mitochondrial mutation rates in a non-linear fashion.

---

## [Author Response]

Reviewer #1:I really liked the idea of comparing synonymous substitution rates and types in closely related species that experience different levels of bedrock radioactivity. I am convinced that there seems to be an association there, and the mutation spectrum results are also interesting, but it was hard to evaluate rival hypotheses for the association without knowing more about the species that live in environments with lower vs higher levels of radioactivity.

One of the most interesting feature of subterranean species belonging to the Asellidae family is their similarities in terms of morphology, lifestyle and life history traits. This high uniformity is likely due to a low rate of phenotypic evolution and a high level of convergence imposed by the subterranean lifestyle. As a result, distinguishing different species requires a high level of expertise and some species cannot be distinguished without molecular tools (Morvan et al., 2013). The birth of the Asellidae family is estimated at -350 My (Morvan et al., 2013) and divergence times between species are comparable to what is observed between species of mammals with very different biologies. These characteristics allow us to compare species that are divergent enough to compute an accurate rate of molecular evolution but which conserved very similar traits. We have included a paragraph in the Materials and methods describing these aspects of the biological model.

However, even if no phenotypic differences were observed for species living in radioactive habitat, we agree that radioactive environments could potentially influence life history traits, growth rate or population size. We have now included a paragraph to discuss these possible confounding factors in the manuscript.

I was also curious how diverged they are at the nucleotide level.

In the manuscript we used the estimation taken from the software CoEvol which gives a dS divided by a relative time (that we noted dS/ra). In the Supplementary file 1, we reported classical measures of the dS (as computed by mapNH and CodeML) which is the measure of the synonymous divergence. In the Figure 1—figure supplement 1, we added branch length values which represent the average nucleotide divergence for all the positions. On average, we found a 4,2% nuclear synonymous divergence between two closely related species (39% for the mitochondrial genome).

Given that the authors have the data to look at these questions, it would be helpful to know how diversity levels and the allele frequency spectrum differs between different species compare for instance.

Levels of genetic diversity (πS or πN) are significantly negatively correlated with the different estimates of natural radioactivity (pGLS p.value < 0.05). This observation is however not informative regarding the question of the mutational impact of natural radioactivity, because nucleotide diversity depends not only on the mutation rate but also on the effective population size (πS=4*Ne*μ). We therefore did not mention this point in the manuscript. We however included estimates of ⲡN and ⲡS in the Supplementary file 1, so that interested readers can have this data.

I also didn't quite know what to make of the quantitative estimates when I expect the level of bedrock activity likely changed quite a bit over the o(Ne) generations of the species. Could the authors comment on that?

We agree this point should be discussed. We included a new paragraph in the Materials and methods justifying this assumption. Natural radioactivity is unlikely to have changed in the time-frame of this study, but species exposure has probably changed due to dispersal. However, population movements should blur a signal, not generate one.

For the mutation spectrum analysis, I think the authors should use a forward variable selection approach, such as the one employed in Harris and Pritchard, 2017.

We agree that performing 6*3 pGLS tests does not take into account the dependency between the different proportions of mutation. At the same time, the test proposed by Harris and Pritchard, 2017, does not take into account phylogenetic inertia. Integrating the phylogenetic signal is important because this is one of the main factors explaining the mutational spectrum variations. To take into account these two types of non-independency in the mutational spectrum analysis and reconcile the two approaches, we first performed the test proposed by Harris and Pritchard, 2017, on the mutation counts as advised by the reviewer. This test gives only one mutation significantly linked to radioactivity : C:G -> A:T. We then performed a pGLS test where p.values were corrected with Holm’s method to adjust for the multiple tests and to control the false discovery rate. In the manuscript we reported the results of this statistical procedure in Table 2 and in the text, as well as in the Materials and methods.

In terms of presentation, I found the Introduction somewhat unhelpful, in that it lumps references for all sorts of ionizing radiation and UV, on both point mutations and microsatellites. For this reader at least, it would be helpful to overview what types have been studied, and what is known specifically for the type examined here. Also this study should be discussed: https://www.ncbi.nlm.nih.gov/pubmed/25809527. On a related but more minor note, I was unconvinced by the argument that this question cannot be studied experimentally and I don't think the argument is needed to motivate their approach.

We agree that a longer Introduction will be helpful and have included 3 new paragraphs or sections of paragraphs to the Introduction. A new paragraph summarizes the different types of mutations generated by radioactivity. We distinguished point mutations from deletions and chromosomal rearrangements, while acknowledging that the current knowledge comes from studies of radiation doses much higher than natural radioactivity. We added the article pointed by the reviewer which was completely relevant in this part as the authors showed that radioactivity generated clustered mutations and deletions. This new introductory paragraph concludes that point mutations have been poorly studied compared to double-strand-breaks or deletions.

We then added a part explaining more extensively what we know about the impact of very low doses of radioactivity such as in the case of natural radioactivity, and which factors could be at play to explain the difficulty to characterize a dose-response relationship.

Finally, we agree that the fact that the long term effect of low doses of radiation is challenging to test experimentally is not needed to motivate our approach. Our goal here is to highlight that a phylogenetic approach has never been used before to tackle that challenging question and can be a complementary approach. We have re-formulated this idea.

In turn, the discussion of life history trait effects on mutation rates in metazoans was both too strong and oddly referenced. As examples, Martin and Palumbi, 1993 is actually an argument for consideration the effect of metabolic rates; Saclier et al., 2018 is only in isopods, when the claim is made for metazoans etc… Similar to my comment on the Introduction, I think a more systematic discussion of the literature is needed.

We agree that this paragraph could be read as a too strong and too general about the relative influence of radioactivity compared to life history traits. As the literature do not allow to extend our results to metazoans, we modified the section to make it clear that this is an observation that may only be valid for isopods and needs to be investigated in other groups.

Finally, I thought the authors could do more to link their findings to studies in other organisms, in particular humans, where there are a number of studies of mutation patterns in populations living in diverse environments (e.g., https://journals.plos.org/plosgenetics/article?id=10.1371/journal.pgen.1006581).

We agree that the mutation spectrum was too rapidly discussed. We have included a new paragraph where we discuss the factors known to modify the mutational spectrum in diverse organisms (mostly human and mice). We also enriched this discussion with studies about other carcinogens known to also modify the mutational spectrum.

Reviewer #2:This manuscript is proposing an attractive model suggesting that natural radioactivity (even at a low level) affects the mutation rate in waterlice. The authors collected unique biological material in a number of areas with variable radioactivity levels. The data and the proposed model are certainly of great interest. However, I find that the manuscript would benefit from a few additional statistical analyses of the data and from softening the discussion about ROS.1) The analysis makes a major assumption that the habitat and the corresponding level of radiation remains unchanged after speciation. This assumption warrants a discussion. Overall, I agree that a violation of this assumption would result in a conservative estimate and main results should not be affected.

This is a point that has also been raised by reviewer #1, please refer to the answer to his fourth comment.

2) I think that the study lacks a test for robustness of pGLS.

Indeed, pGLS makes some assumptions for which robustness should be tested. pGLS supposes normality of residuals, homoscedasticity, absence of influential cases, an evolutionary process (here a brownian motion) which is homogenous across the branches of the phylogenetic tree. Normality was checked by plotting residuals of models. Homoscedasticity was checked by plotting the residuals against the fitted values of models. Absence of influential cases was tested by a jackknife method consisting in removing one by one each species of the data and redoing the test each time the pGLS test p-value was below 0.05. Robustness to the evolutionary model was tested by doing the pGLS with the four main evolutionary models (Blomberg, Martins, Pagel and Brownian). P.values remained below 0.05 whatever the model used. These additional tests have been added in the Statistical analyses part. Testing robustness to the homogeneity of the model of evolution across the tree is more complicated. A test proposed by Mazel et al., 2016, Ecology incorporates shifts in the parameters of the model of evolution along the tree. However, the function behind this test needs a minimum of 64 taxa to be powerful enough in detecting shifts (Eastman et al., 2011, Evolution). Despite this, we performed Mazel’s test but detected no shift. For the sake of clarity and to keep a simple and straightforward presentation of the results, we have not included this latest additional result.

3) The manuscript should discuss potential covariates and confounders. Is it possible that variables other than radioactivity level are responsible for the signal?

This is a major point that has also been raised by the reviewer #1, please refer to the answer to his question n°1.

4) The conclusions of the manuscript are based on Figure 2 showing non-ulotrametric trees and ratios of synonymous branch lengths. It would be great to show the data for "polymorphic" variants (just terminal branches).

This is a point which converge with the question asked by the reviewer #1, please refer to the answer to his question n°2.

5) In human cells, radiation primarily induces deletions associated with the microhomology repair. If this is also the case in the waterlice system, the authors should be able to find a highly specific and strong signal on top of the results about point mutations. This would dramatically strengthen the conclusions.

Indeed, the major effect of radioactivity on DNA is to generate deletions (of note, we precise this point in the Introduction). However, the dataset used in this study was generated using transcriptome sequencing, not genome sequencing, and it is therefore limited to protein coding sequences. It is very unlikely to find deletions in protein coding sequences as they are often deleterious and removed by natural selection. Studying the impact of radioactivity on the deletion rate should be done on the non-coding part of the genome, which we don’t have access to for the time being.

Counting the number of deletion polymorphisms in the coding sequences is nevertheless possible. We used Freebayes to count the number of polymorphic deletions in the 2490 1-to-1 orthologous genes. We find no correlation between the level of radioactivity and the number of deletions, whatever the measure used (e.g. pGLS, p.value=0.7034 for λ15). But again this is not a valid demonstration that natural radioactivity has no influence on the deletion rate.

6) I do not find that the shift in mutation spectra is a sufficiently strong evidence in favor of ROS. This either requires a much stronger argument, or the discussion about the role of oxidative damage should be softened.

We agree and have included a completely new paragraph to discuss the evidence supporting a potential influence of the ROS. We have softened our conclusion by saying that the modification of the mutational spectrum suggests an impact of radioactivity mediated by the formation of ROS.

Reviewer #3:In this study, Saclier et al. explore the effects of natural radioactivity on the germline mutation rate and spectra of subterranean waterlice, aiming to address the question of whether radioactive habitats have long-term evolutionary effects on the genomes of species which inhabit them. The use of this organism to test such a fundamental question in evolutionary biology is very clever, as waterlice are not exposed to confounding effects of UV radiation and have naturally limited dispersal. I found the paper well-written and a fun topic to read, and the arguments that natural radioactivity can have a tangible impact on genome evolution were compelling.I have a few concerns and suggestions about the regression models and statistical analyses.Figure 2: This would be easier to follow if points were labeled/colored/shaped to indicate the corresponding species, site, and bedrock formation type.

The Figure 2 has been modified following these recommendations.

Table 1: The authors use independent regression models to test for the associations between mutation rate and each three variables of interest: α radioactivity, Received Dose, and λ-15. Why did the authors perform three separate simple regressions rather than a multiple regression using all three of these variables as covariates? Given that λ-15 and Received Dose appear to be stronger predictors of mutation rate than α radioactivity, does this mean the statistically significant association of α radioactivity goes away when adjusting for the other two variables? I expect these three variables to be highly collinear and too many covariates could easily result in overfitting with such a small sample, but the claims drawn from this table would benefit from a more nuanced statistical analysis.

We chose to do three separate tests for three main reasons. First, the λ15 measure allows us to include the two sites nearby mines (showing a disequilibrium in the uranium's chains). Doing a single test with the three measures means we have to remove these two locations. Second, as pointed by the reviewer, as we have only 14 points, a test with 3 predictors will lack power to detect an effect. Finally, and maybe most importantly, the three measures of radioactivity are redundant. Absence of collinearity among predictors is required for the validity of a linear model. Conclusions about the impact of multiple predictors that are collinear are unreliable in the linear model framework (Quinn and Keough, 2002). α radioactivity is a part of the total radioactivity which is measured to compute the received dose of radioactivity, thus the two measures are expected to be collinear (R² = 0.82). λ15 is also collinear with the two other measures (R² = 0.61 and 0.77, with α radio and Received dose, respectively). As a result, if we do a single pGLS with the synonymous substitution rate against the three measures of radioactivity, none of the three factors remain significant (p.values = 0.29, 0.76 and 0.08 for α radioactivity, Received dose and λ15, respectively). Absence of statistical significance in such a test is the result of collinearity and falsely suggests an absence of effect. For the sake of clarity, we have not included this test in the manuscript, however we added a sentence in the caption of Table 1 justifying the absence of such test because of collinearity.

Also, were there any additional environmental covariates that may have been measured when samples were collected or integrated from other data sources (e.g., latitude, distance to nearest industrial activities, etc.)? If possible, it would be interesting to see if other environmental factors can also explain the variation in mutation rates, but I recognize that such data may not be available, and even if it were, this analysis may not be feasible with a small sample.

We agree that a more comprehensive description of other factors, in particular potentially mutagenic factors, would be valuable. Unfortunately, the already complex sampling strategy of pairs of contrasted species associated with a detailed description of the natural radioactivity (in situ characterisation and received dose modelisation) was already a challenge to put in place in the field and we could not reasonably complexify it even more.

We have modified the Discussion to integrate a description of other potential confounding factors. Of particular interest are heavy metals whose concentration is correlated with radioactivity as they are a sub-product of the decay chains. To disentangle the impact of radioactivity and heavy metal, it would require a lot of observations, which does not correspond to the sampling strategy of this study.

Table 2: Performing 3 separate regressions for each of 6 mutation types results in some multiple testing issues that must be addressed, at a conservative Bonferroni-corrected α value of.05/18=.0028, only the C:G>A>T mutation class is statistically significant across all 3 explanatory variables, but the A:T>T:A class is not. This multiple testing burden could perhaps be alleviated by using multiple regression as suggested above. Further, the response variables of the 6 regressions are proportions that add to 1, so these are not independent statistical tests, even though they are analyzed and presented as such. Is there a different statistical model that can be used that takes into account the interdependence of the 6 components of the mutation spectrum?

As suggested by the reviewer #1, we performed the test proposed by Harris and Pritchard, 2017, that we coupled with a pGLS where p.values have been corrected for multiple hypothesis testing following Holm’s method. Please refer to the answer to question #5 of the Reviewer #1.

How well-correlated are the nuclear and mitochondrial dS/ra values across samples? A priori, I expect them to be strongly correlated, but it would be interesting and straightforward to investigate if radioactivity levels also affected relative differences in nuclear and mitochondrial mutation rates, e.g., perhaps radioactivity increases nuclear mutation rates in a linear fashion, but mitochondrial mutation rates in a non-linear fashion.

Indeed, the mitochondrial and the nuclear dS are well correlated (PGLS, p.value = 0.0065, Cox-Snell pseudoR² = 0.41). However the rate ratio [mitochondrial dS/nuclear dS] is not linearly correlated with the level of radioactivity (p.values > 0.05) even if a positive trend is found with α radioactivity (p.value=0.09). A graphical evaluation suggests that for exposure to low levels of natural radioactivity (<3 µSv/h), the ratio seems to increase linearly, suggesting a faster acceleration of the mitochondrial rate, but for higher doses this ratio seems to drop. However, with only 14 points, this pattern cannot be accurately described. One complication is the saturation of the mitochondrial rate. As the mitochondrial rate is about 10X times higher than the nuclear one, an important rate acceleration will tend to be less well estimated in the mitochondrial genome compared to the nuclear one. When working on rate ratios, this bias alone could generate a pattern. A much bigger dataset and methods to compensate for saturation are therefore necessary to characterize the relative differences in nuclear and mitochondrial mutation rates in response to natural radioactivity. For the sake of clarity, we have not included these new analyses.